# Machine learning forecasts for seasonal epidemic peaks: Lessons learnt from an atypical respiratory syncytial virus season

**Roger A. Morbey**[1]*, **Daniel Todkill**[1], **Conall Watson**[2], **Alex J. Elliot**[1]

**1** Real-Time Syndromic Surveillance Team, Field Services, Health Protection Operations, UK Health Security Agency, Birmingham, United Kingdom, **2** Immunisation and Vaccine Preventable Diseases Division, UK Health Security Agency, London, United Kingdom

* roger.morbey@ukhsa.gov.uk

**Data Availability Statement:** https://www.gov.uk/government/collections/syndromic-surveillance-systems-and-analyses

## Abstract

Seasonal peaks in infectious disease incidence put pressures on health services. Therefore, early warning of the timing and magnitude of peak activity during seasonal epidemics can provide information for public health practitioners to take appropriate action. Whilst many infectious diseases have predictable seasonality, newly emerging diseases and the impact of public health interventions can result in unprecedented seasonal activity. We propose a Machine Learning process for generating short-term forecasts, where models are selected based on their ability to correctly forecast peaks in activity, and can be useful during atypical seasons. We have validated our forecasts using typical and atypical seasonal activity, using respiratory syncytial virus (RSV) activity during 2019–2021 as an example. During the winter of 2020/21 the usual winter peak in RSV activity in England did not occur but was 'deferred' until the Spring of 2021. We compare a range of Machine Learning regression models, with alternate models including different independent variables, e.g. with or without seasonality or trend variables. We show that the best-fitting model which minimises daily forecast errors is not the best model for forecasting peaks when the selection criterion is based on peak timing and magnitude. Furthermore, we show that best-fitting models for typical seasons contain different variables to those for atypical seasons. Specifically, including seasonality in models improves performance during typical seasons but worsens it for the atypical seasons.

## Introduction

Many respiratory and gastrointestinal infectious diseases have a seasonal component, resulting in annual peaks in disease incidence. These seasonal epidemics create an additional and significant strain on health services through increased ED visits, GP consultations and hospital admissions, and may require public health interventions to mitigate their effects [1, 2]. Whilst typical seasonal activity can be modelled using historical data there is often variation in the timing and intensity (i.e. maximum number of cases) of annual peaks [3]. Therefore, accurate

**Funding:** The author(s) received no specific funding for this work.

**Competing interests:** The authors have declared that no competing interests exist.

short-term forecasts for the timing and intensity of seasonal peaks would provide very useful information for public health decision makers.

There is substantial literature on forecasting, particularly for influenza [4–11]. However, forecast models are usually assessed on whether they can detect increased activity associated with outbreaks or the accuracy of daily or weekly forecasts, not the accuracy of forecasting peak activity [12]. Model selection methods that minimise forecast errors or maximise the sensitivity and specificity of outbreak detection will not necessarily provide models that are optimised for forecasting the timing and intensity of seasonal peaks. Therefore, we have developed a selection criterion based on the accuracy of forecasting peaks. By contrast, selection criteria that gives equal weight to all forecast errors in the training data, may perform well for most of the year but not around the crucial period of an annual peak.

One key motivation for public health surveillance is that infectious diseases do not always follow historical seasonal patterns. Emerging diseases can result in dramatic 'out of season' increases in healthcare use activity, like the 2009 H1N1 influenza pandemic and the Sars-Cov-2 Covid-19 pandemic. Also, major interventions like the introduction of new vaccines or national lockdowns can change the seasonality of diseases in unpredictable ways. Thus, when activity diverges from seasonal norms and comparison with previous years is no longer informative, real-time forecasts of peaks are even more important.

A model that is trained solely on historical data with consistent seasonality might be considered as being 'overfitted' to a specific seasonal pattern and perform badly when atypical activity occurs. However, we wish to design forecasts that perform well, even when seasonal activity is unprecedented, we define these as 'black swan' seasons after Nassim Taleb's book, "The Black Swan" [13] about rare and unpredictable events. Therefore, we have validated our models using black swan seasons, comparing models trained with and without a seasonal component.

Respiratory syncytial virus (RSV) has a major impact on health, particularly on young children and the elderly and is an example of a seasonal respiratory disease. In temperate countries such as the United Kingdom, RSV activity typically peaks in December [14, 15], which has been consistently monitored for several decades [16]. However, recent years have provided examples of black swan seasons for RSV. Firstly, during the winter of 2020/21 there was no usual seasonal increase and peak in activity for RSV. Second, there was a 'deferred' out-of-season peak in RSV during the spring/summer of 2021 [17]. The most likely cause for this change in seasonality was the introduction of national lock-down measures during the Covid-19 pandemic which changed behaviours, thus reducing transmission during the winter of 2020/21 [18–20].

RSV activity is monitored by the UK Health Security Agency (UKHSA) using both laboratory surveillance and real-time syndromic surveillance [21]. Syndromic surveillance involves monitoring health care diagnostic data that is available earlier than laboratory results. Thus, UKHSA syndromic surveillance data can be used to provide daily forecasts that could give early warning of peak activity [22]. We validate our approach using the black swan seasons for RSV of 2020 and 2021.

In this paper we present a method for generating real-time short-term forecasts for the timing and intensity (or "height") of seasonal epidemic peaks. We create a novel measure for selecting models, based on their specific accuracy in forecasting peaks. Furthermore, we use real examples of RSV black swan seasons to validate whether models trained on historical data perform better or worse when seasonality is included as a factor. The methods presented, use machine learning techniques to create automated pipelines for generating forecasts, therefore methods are highly generalisable and quick to implement in existing surveillance systems.

## Methods

### Overview

As a pilot example for our forecasting method, we use the number of calls to a national telephone health helpline (NHS 111) for cough in children aged under 5 years as a syndromic indicator for RSV [23], since 2013. This data is anonymised and used for routine surveillance, with the surveillance outputs published weekly on Remote health advice: weekly bulletins for 2023 - GOV.UK (www.gov.uk).

We used a machine learning approach to create reproducible pipelines for generating forecasts. The approach included the following stages: formatting and splitting the data into 'train' and 'test' sets, training alternate models, creating peak forecasts, and validating the models. Each stage in the process is described in more detail below. The machine learning approach meant many alternative models could be compared concurrently, with consistency assured by using the same pipeline for training, testing, and validating models.

### Formatting and splitting the data

Raw data was extracted as daily counts and then smoothed to remove day of the week and holiday effects and reveal the underlying epidemic trend. Our forecasting approach was to estimate the future development of an epidemic curve by considering where we currently are on the epidemic curve. Therefore, predictor variables included the current slope of the curve and current daily count, which we define in this paper as 'intensity'. All models included the change in counts between the most recent two data points and the second order difference, to estimate both current trend and the current rate of change in that trend. The process for formatting data included normalising all variables prior to training models. Historical data was randomly split into a training and test data set. 80% of the data was used to train the models and the remaining 20% was used to independently test and compare the models.

### Training alternate models

We tested a range of popular machine learning methods for our models and further expanded the number of models by including several variations. The regression learners we used for our models included; linear regression, generalised linear models with elastic net regularization (with and without internal optimisation of parameter lambda), k-Nearest-Neighbour regression, Kriging regression, random regression forest, support vector machine for regression, and eXtreme Gradient Boosting regression. Each of these eight different regression methods were applied with each of the variants described below.

Three different approaches to modelling seasonality were tested: firstly, with no seasonal predictors, secondly with binary variables for months of the year and thirdly using Fourier transformations to model annual seasonality using two sin and two cos terms. Similarly, three different variants were used for modelling longer term trends: no trend, a linear trend, and a quadratic trend. To model more complex relationships than just linear between current intensity and forecast intensity, a variant was included in half the models with a quadratic term for intensity. Finally, in case single day spikes in activity disproportionately affected forecasts, a variant was included that used an average of the past three days, rather than just the most recent activity. Combining the different regression methods and the variants, gave 288 alternate models to be tested.

## Creating peak forecasts

The datasets were labelled using the actual activity for the next 28 days as the targets that we were trying to forecast. Thus, each of the 288 alternate models were trained separately to forecast 1 day, 2 days, etc up to four weeks ahead. For each date and model, the highest of the 28 forecasts was then used as the forecast peak to be compared with the actual peak in activity over the 28 days.

The model that is best at predicting one day ahead may not be the best for predicting further ahead. Therefore, we created an alternative ensemble peak forecast model that used different models for forecasting one day ahead and for longer lead times. We also created a third alternate peak forecast model that combined weighted forecasts for different forecast leads, i.e. the forecast for tomorrow's activity, used a combination of the 1 day ahead forecast created today, the two day ahead forecast created yesterday, etc. The weighting for these weighted ensemble forecasts was based on the comparative accuracy of different lead times, so more weight was given to the one day ahead forecast which was more accurate than the 28 day ahead forecast etc.

The accuracy of daily forecasts can be easily measured by considering the "forecast errors", i.e. the absolute difference between daily forecasts and the actual labelled data. However, the accuracy of forecasting the peak in activity across 28 days is more complex. Firstly, we want the peak forecast to perform well in two dimensions, timing and intensity. Secondly, we want to ensure that models perform as well as possible during seasonal peaks, with the timeliness of peak forecasts being less important when intensity is low across the whole 28 days. Therefore, we created a 'peak error' measure that gives a score between 0 and 1 to all peak forecasts, considering timing and intensity. The peak error measure can be described using the following equation:

$$y_d = \frac{\max\,(f_d, x_d)}{\max\,(x)} \;*\; \left(\left(1 + \frac{t_d}{27}\right) \;*\; \left(1 + \frac{i_d}{\max\,(i)}\right) - 1\right)/3$$

Where $y_d$ is the peak error on day $d$, $x_d$ is the actual smoothed count on day $d$, $f_d$ is the forecast peak intensity on day $d$, $\max(x)$ is the maximum of all actual and predicted smoothed counts, $t_d$ is the difference in days between the date of the peak forecast and when the actual peak occurred, $i_d$ is the difference between the peak forecast's intensity and the actual peak, and $\max(i)$ is the maximum error seen in predicting forecast intensity. Our peak error measure is zero if the peak forecast correctly identifies both the date and intensity of the peak. The measure increases as the difference between peak forecast intensity and actual peak intensity increases. Also, the measure increases as the difference between the forecast date and actual date of peak increases, but this increase is less if both actual and forecast activity is low. Table 1 illustrates how this measure would score illustrative examples of counts and forecasts.

## RSV example

For our pilot example we used the syndromic indicator, NHS 111 daily cough calls for children aged under 5 years in England. This indicator has previously been shown to be closely correlated to outbreaks of RSV [24]. We used anonymised data that was provided for public health surveillance since the start of the NHS 111 service on 28[th] Sept 2013. The models were trained and tested on data prior to, 2019–20, i.e. 28/03/13–31/08/19. Forecast models were validated using data from three periods: a typical winter season 01/10/19–15/01/20, and two atypical periods; winter season 01/10/20–15/01/21 and 01/03/21–30/06/21. The second winter season was atypical because there was very little RSV activity, and the Spring of 2021 was unusual because there was a 'deferred' peak in RSV activity. Including these black swan seasons in our

**Table 1. Illustrative example of peak forecasts for NHS 111 cough calls and peak error measures, (counts of cough calls and forecasts range 5–1000).**

| Date of forecast | Actual peak number of calls in next 28 days | Forecast peak | Peak error measure |
|---|---|---|---|
| 30th Nov | 1000 on 28th Dec | 1000 on 1st Dec | 0.333 |
| 30th Nov | 1000 on 28th Dec | 1000 on 15th Dec | 0.160 |
| 30th Nov | 1000 on 28th Dec | 1000 on 28th Dec | 0.000 |
| 30th Nov | 1000 on 28th Dec | 5 on 28th Dec | 0.333 |
| 30th Nov | 1000 on 28th Dec | 500 on 28th Dec | 0.168 |
| 30th Nov | 1000 on 28th Dec | 5 on 1st Dec | 1.000 |
| 30th Nov | 1000 on 28th Dec | 500 on 15th Dec | 0.409 |
| 31st May | 5 on 28th Jun | 5 on 28th Jun | 0.000 |
| 31st May | 5 on 28th Jun | 5 on 1st Jun | 0.002 |
| 31st May | 5 on 28th Jun | 500 on 28th Jun | 0.083 |
| 31st May | 5 on 28th Jun | 1000 on 1st Jun | 1.000 |

validation meant we could check if models were performed well in all seasons, especially when seasonality was included as a model variant.

## Results

Between 28/03/13 and 31/08/19 there were 1,060,624 calls to NHS 111 where the primary diagnosis was cough in a child aged under 5 years. The daily volume had a mean of 489.4 calls, ranging from 52 to 2,609. During the 'typical' winter seasons peak timing varied from 25th November in 2018 to 27th December in 2014 & 2016, whilst peak intensity varied from 1,884 in 2013 to 2,609 in 2014, Table 2. During the winter of 2020/21 the peak was just 409 on 18th November, and between 01/03/21 and 30/06/21 there was a peak of 2,589 on 31st May, Table 2.

During training, problems occurred when trying to fit methods using the Kriging regression method. This method would not converge due to an inability to invert the covariance matrix. Therefore, we used the diagonal inflation method, with a nugget set to 1e-8*variance, to overcome this issue. However, this method was considerably slower to converge than other methods and overall had a lower forecast accuracy, and so was excluded from further analysis.

Absolute forecast errors were calculated for the test data set. The mean of the forecast errors increased monotonically with lead time, so that the overall mean forecast error for next day forecasts was 55.0, whilst for 28 day-ahead forecasts it was 113.6. Models using random forest regression had the lowest mean forecast error of 31.6, with extreme gradient boosting models

**Table 2. Peak intensity and timing of cough calls in children aged less than 5 years by RSV season defined as 1st October-15th January.**

| RSV Season | Date of peak intensity | Peak Intensity |
|---|---|---|
| 13/14 | 21st December | 1,884 |
| 14/15 | 27th December | 2,609 |
| 15/16 | 6th December | 2,251 |
| 16/17 | 27th December | 2,091 |
| 17/18 | 26th November | 2,069 |
| 18/19 | 25th November | 1,942 |
| 19/20 | 1st December | 2,152 |
| 20/21 | 18th November | 409 |
| 1st Mar-30th June 21 | 31st May | 2,589 |

having the highest average forecast errors, 289.6. The best-fitting model, with the lowest overall average forecast error was a random forest regression, with Fourier seasonality, a quadratic trend and using an average of the last 3 data points. The 12 models with the lowest forecast errors were all random forest regressions with Fourier seasonality. When stratifying forecast errors by lead time, the best-fitting models are still random forest regression with Fourier seasonality, except for the 1 day ahead forecasts. The best model for 1 day ahead forecasts, used linear regression with seasonality modelled by a variable for month. S1-S3 Tables in S1 File illustrate the mean peak error by regression type, model and the best-fitting models for each forecast lead.

An ensemble forecast model was created combining the linear regression model with the lowest mean forecast error for 1 day ahead forecasts, and the random forest model that performed best for forecasts more than 1 day ahead. Similarly, a weighted forecast model was created by combining forecasts made on different dates. When validated using the Winter of 2019/20, the ensemble forecast model had a mean peak error of 0.086 and the best weighted forecast model had a peak error of 0.077. However, when compared to non-ensemble models, some of these had lower mean peak errors. Therefore, the ensemble and weighted variants was not pursued further.

When validated against the typical winter season of 2019/20 the model with the lowest mean peak error (0.052) was a generalised linear model (GLM) with elastic net regularization (with internally optimized lambda), incorporating month of year, a quadratic trend and quadratic term for intensity. By comparison, the mean peak errors were higher for the atypical seasons of winter 2020/21 and summer 2021. The model with the lowest mean peak error (0.138) for the winter of 2020/21 used support vector machine regression with a linear trend but no seasonality variables. The model with the lowest mean peak error (0.129) for the summer of 2021 used linear regression with a linear trend, a quadratic term for intensity and averaging over three data points. It was noticeable that the models with the lowest peak errors for the typical season included seasonality variables, whilst those for the atypical seasons did not. Fig 1 shows violin plots for the mean peak error by validation season stratified by seasonality variant. As a sensitivity analysis, Table 3 shows the mean peak errors for each of the three validation seasons, stratified by regression type and model variants. The inclusion of seasonality variables is the single most important factor affecting peak errors.

Models that included a seasonal component, forecast peaks during the Winter 2020/21 season that did not occur. Similarly, seasonal models during the Summer of 2021 predicted that activity would fall to usual summer levels whilst activity was still rising due to the deferred peak. Figs 2–4 show forecasts for two models, using linear regression with a quadratic trend, one model with no seasonal variables the other with Fourier transform coefficients to model seasonality.

## Discussion

Here, we have used a machine learning approach to train models to forecast seasonal epidemics, comparing different regression methods and model variants. Interestingly, the models with the lowest daily forecast errors, i.e. differences between daily forecasts and actual counts, were not the models that were best at predicting the timing and intensity of a seasonal peak. The models with the lowest forecast errors were random forest regressions with Fourier seasonality, although the best model for one-day ahead forecasts used linear regression with months to model seasonality. The best model for predicting the timing and intensity of the 2019/20 winter peak was a generalised linear model with elastic net regularization, incorporating month, a quadratic trend and a quadratic term for intensity. When validating using

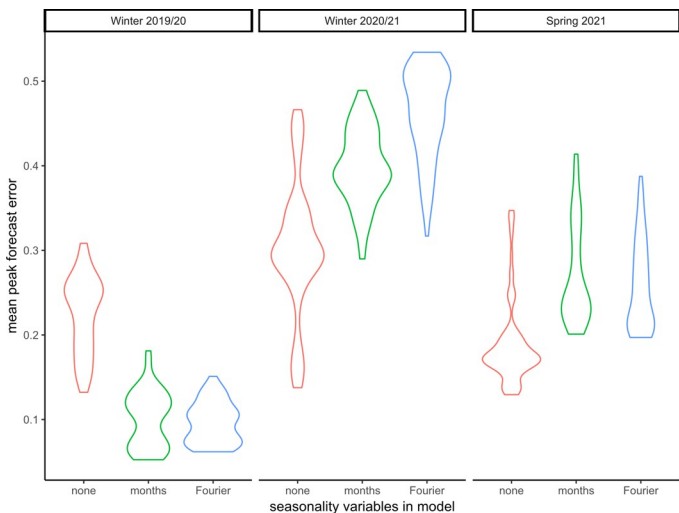

**Fig 1. Violin plot showing the density curves (width is approximate frequency of data points) of mean peak error by validation season, stratified by seasonality variant.**

atypical seasons the best models for predicting peak timing and intensity did not include any seasonality variables.

We developed a new peak error measure to validate models based on their ability to correctly forecast the timing and intensity of peak activity. Models that were optimal in terms of daily forecast errors, were not the same as those selected based on the peak error measure. One model may outperform a rival for most of the year when there is no epidemic and consequently have a lower mean forecast error, however, if it performs worse around an annual peak it will score worse under our measure. Our peak error measure will score a model poorly if it misses a seasonal peak or predicts a peak when one does not occur. We found that random forest regression models had the lowest forecast errors but were outperformed by GLM with elastic net regularization and by linear regression in terms of peak errors. Therefore, it is important that model selection is not done after the calculation of forecasts, but at the later stage after constructing peak forecasts.

It is sometimes argued that machine learning models are more objective than theory-based models because they are trained solely on the data, without any assumptions from the modeller about dynamics or causality. However, relying solely on historical data is a weakness when unprecedented or black swan events occur. We have illustrated the problem of black swan seasons using the example of RSV. If our forecast models were selected prior to 2020, the best forecast models would have included seasonality as RSV had a consistent single annual peak towards the end of each year. However, these forecast models would have performed poorly during 2020 and 2021. With hindsight we can see that a better approach would have been to either not include seasonality in our models, or to include a term that rescales the importance of seasonality, to allow for the possibility that peaks could occur at different times of year to those previously seen. This is an important insight into the dangers of 'overfitting' models based solely on historical data. When we have reason to believe that unprecedented events could occur, we need to avoid including variables that constrain our models to behave as if the past includes all possibilities. This is particularly true in epidemiology where emerging diseases and climate change mean that the future is going to include more unprecedented events.

**Table 3. Mean peak error by validation season, stratified by regression type and model variant.**

| | Validation season | | | All seasons |
|---|---|---|---|---|
| | Winter 2019/20 | Winter 2020/21 | Summer 2021 | |
| **All models** | 0.141 | 0.388 | 0.239 | 0.256 |
| **Regression type** | | | | |
| Generalised linear models with elastic net regularization (with internally optimized lambda) | 0.120 | 0.394 | 0.195 | 0.236 |
| Linear | 0.116 | 0.372 | 0.223 | 0.237 |
| Random Forest | 0.132 | 0.359 | 0.229 | 0.240 |
| Generalised linear models with elastic net regularization (without internally optimized lambda) | 0.138 | 0.398 | 0.201 | 0.246 |
| Support vector machines | 0.174 | 0.331 | 0.305 | 0.270 |
| Extreme gradient boosting | 0.144 | 0.448 | 0.221 | 0.271 |
| k-nearest-neighbour | 0.160 | 0.415 | 0.301 | 0.292 |
| **Seasonality variables included** | | | | |
| None | 0.225 | 0.302 | 0.188 | 0.239 |
| months | 0.101 | 0.398 | 0.272 | 0.257 |
| Fourier | 0.096 | 0.463 | 0.258 | 0.273 |
| **Trend variables included** | | | | |
| None | 0.133 | 0.383 | 0.232 | 0.250 |
| Linear | 0.141 | 0.392 | 0.236 | 0.256 |
| quadratic | 0.148 | 0.388 | 0.251 | 0.262 |
| **Quadratic term for intensity included** | | | | |
| Yes | 0.137 | 0.381 | 0.236 | 0.251 |
| No | 0.144 | 0.394 | 0.243 | 0.261 |
| **Forecast based on single day or average of three days** | | | | |
| Three days | 0.140 | 0.387 | 0.239 | 0.255 |
| Single day | 0.141 | 0.389 | 0.240 | 0.257 |

However, to accept this principle means on occasion selecting models that are not the best fit to our data.

One method developed specifically because the timing of seasonal epidemics varies is the Moving Epidemic Method (MEM) [3]. MEM is used across Europe and in many other countries as a standard way to assess the onset of the influenza season and its current intensity, although it is not a forecasting tool.

Methods of model selection vary depending on the forecasting approach, however an accepted minimum standard is that models should be validated using real data that was not used in training the forecasts [25]. For example, Zarebski et al, describe a method of selecting between mechanistic models which use a Bayes factor approach to show which model best fits the actual influenza epidemic forecast [26]. Moss et al, used a Bayesian approach to model seasonal influenza epidemics in Australia and integrate them into public health practice [8]. They acknowledge that forecast uncertainty can be reduced by assuming that seasons will stay within expected parameters, and initially calibrated their models to reflect the duration, timing, and intensity of previous seasons. However, when the 2017 season was outside of their model parameters, exceeding both historical data and expert estimates, then re-calibration was necessary. The Centers for Disease Control and Prevention host an annual influenza season forecasting challenge [11]. The 2015–16 challenge used separate metrics for assessing models' ability to predict the onset week, peak week, and peak intensity of seasonal influenza. Whilst they did not consider out-of-season influenza they did note that forecasts were worse where peak timing and intensity were atypical. Importantly, forecasts that are trained solely on winter data, have not been validated for predicting atypical peaks at other times of year.

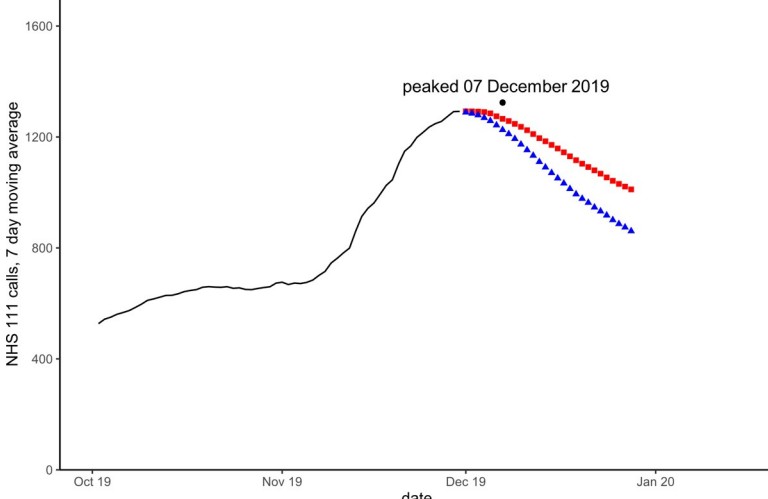

**Fig 2. Forecasts for NHS 111 cough calls in under 5 years on 30 November 2019.** Blue triangles are forecast with modelled seasonality, red squares without seasonality. The black dot and date refer to the actual observed peak of activity during the period.

We have deliberately tried to adopt a simple, generalisable, easy to apply approach that can be extended to other seasonal epidemics, either respiratory or gastrointestinal. Therefore, we have not attempted to model specific disease characteristics and our forecasts are unlikely to be as accurate as transmission models which may consider many factors, such as vaccine effectiveness or weather variables. Also, we have focussed on syndromic data, which may predict pressures on health care systems but does not necessarily align exactly with community incidence of a specific disease. Thus, an emerging respiratory disease may produce similar symptoms to influenza or RSV seasons used to train a forecast model, but the epidemic curve may well have new characteristics. Furthermore, our approach to forecasting is based on short-

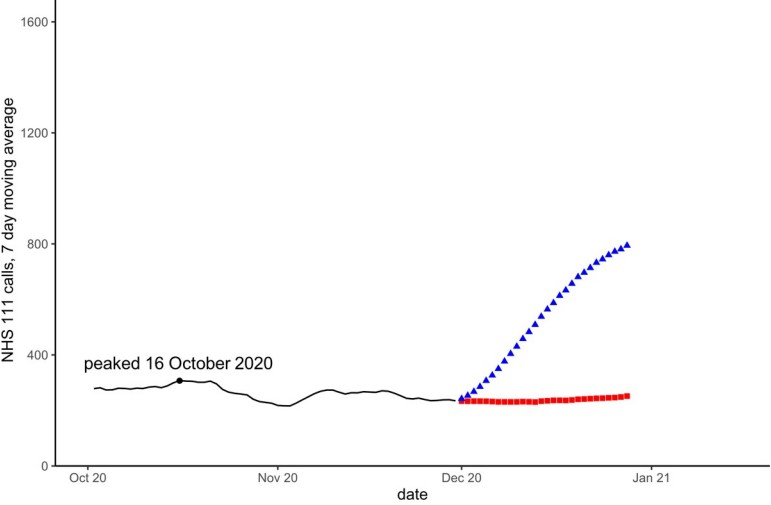

**Fig 3. Forecasts for NHS 111 cough calls in under 5 years on 30 November 2020.** Blue triangles are forecast with modelled seasonality, red squares without seasonality. The black dot and date refer to the actual observed peak of activity during the period.

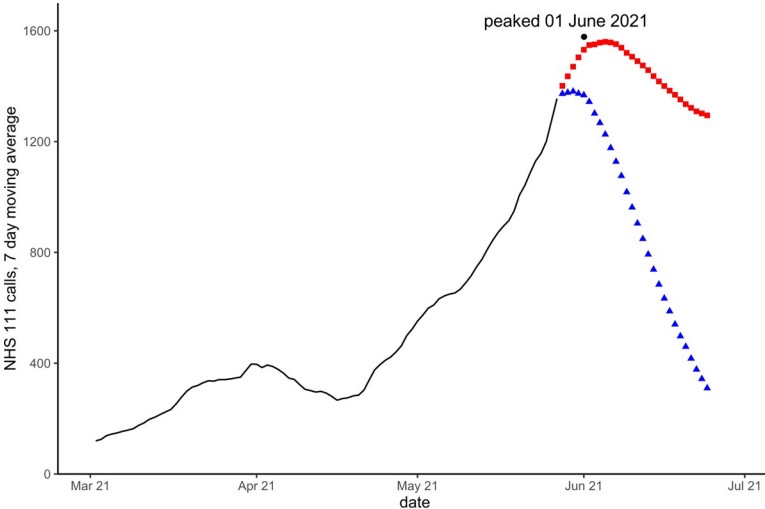

**Fig 4. Forecasts for NHS 111 cough calls in under 5 years on 27 May 2021.** Blue triangles are forecast with modelled seasonality, red squares without seasonality. The black dot and date refer to the actual observed peak of activity during the period.

term forecasts made in real-time, therefore we can only realistically provide a short window of early warning of peak activity once an epidemic has already started. We have not attempted the much harder task of predicting the onset of an epidemic before it has started.

In practice, the utility of our approach will depend on the usefulness of our forecasts for public health decision makers during seasonal and atypical epidemics. Therefore, we propose applying our example model as a pilot for RSV surveillance within England, and if successful extending to influenza and COVID-19 surveillance. Further work, to evaluate the utility of our approach will also require developing methods to communicate the uncertainty around estimates for peak intensity and timing to users.

In conclusion, we have developed a process for training and selecting forecast models that can be applied to real-time public health surveillance data. We have developed a new selection criterion based on the peak error measure which specifically chooses models that are best at identifying the timing and intensity of seasonal epidemics. Furthermore, we have demonstrated that although model fit can be improved by modelling seasonality this can result in over-fitting when unprecedented black swan seasons occur.

## Supporting information

**S1 File.**
(DOCX)

## Acknowledgments

We acknowledge the UK Health Security Agency Real-time Syndromic Surveillance Team for technical expertise in delivering the daily syndromic service. We also thank syndromic data providers: NHS 111 and NHS England.

## Author Contributions

**Conceptualization:** Roger A. Morbey.

**Data curation:** Roger A. Morbey.

**Formal analysis:** Roger A. Morbey.

**Investigation:** Roger A. Morbey.

**Methodology:** Roger A. Morbey.

**Project administration:** Roger A. Morbey.

**Validation:** Roger A. Morbey, Conall Watson.

**Writing – original draft:** Roger A. Morbey.

**Writing – review & editing:** Roger A. Morbey, Daniel Todkill, Conall Watson, Alex J. Elliot.

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
