## [Decision Letter · Decision Letter 0]

14 Jul 2023

PONE-D-22-35171Machine learning forecasts for seasonal epidemic peaks: ‘Black swan seasons’ and the dangers of overfitting seasonalityPLOS ONE

Dear Dr. Morbey,

Thank you for submitting your manuscript to PLOS ONE. After careful consideration, we feel that it has merit but does not fully meet PLOS ONE’s publication criteria as it currently stands. Therefore, we invite you to submit a revised version of the manuscript that addresses the points raised during the review process.

We look forward to receiving your revised manuscript.

Kind regards,

André Ricardo Ribas Freitas

Academic Editor

PLOS ONE

Journal Requirements:

2. In your Methods section, please confirm that all data sources you used were publicly available and anonymized. If this is not the case, please provide information on what permissions you were granted to access these data.

R.M., and A.J.E. receive support from the National Institute for Health Research (NIHR) Health Protection Research Unit (HPRU) in Emergency Preparedness and Response. A.J.E. receives support from the NIHR HPRU in Gastrointestinal Infections. The views expressed are those of the author(s) and not necessarily those of the NIHR, UK Health Security Agency or the Department of Health and Social Care.

Additional Editor Comments:

We thank the authors of this manuscript for their interest in publishing in our journal.

The submitted work has several innovative aspects, however some issues were pointed out by our reviewer. I emphasize that the reviewer raised questions associated with the use of the term “Black swan seasons”, as we had already warned in previous correspondence.

We hope that a new version will be submitted so that we can evaluate and publish it.

Reviewers' comments:

Reviewer's Responses to Questions

**Comments to the Author**

1. Is the manuscript technically sound, and do the data support the conclusions?

Reviewer #1: Yes

2. Has the statistical analysis been performed appropriately and rigorously? 

Reviewer #1: Yes

3. Have the authors made all data underlying the findings in their manuscript fully available?

Reviewer #1: No

4. Is the manuscript presented in an intelligible fashion and written in standard English?

Reviewer #1: Yes

5. Review Comments to the Author

Reviewer #1: Summary

• The authors explore different types of models that forecast epidemic peaks such as of RSV during typical RSV season and out-of-season (black swan) which may have been driven by COVID-19 stringency measures that also reduced contacts among individuals during typical RSV season. They train different models (with or without different terms including seasonality) and test their performance on test data which include that during 2020/2021 RSV season and its delayed epidemic, and compute and compare forecasts errors to identify models that perform better. Overall, I think the paper is well written and could be considered for publication after minor collections.

• A major point is my alternative idea to authors’ line of reasoning; I think seasonality is intrinsic to RSV dynamics or any kind of seasonal respiratory infection and suggesting that models without seasonality could perform better during black swan seasons may be misleading (though true). An alternative approach would be to still maintain the seasonal term in RSV models while including a term that alters normal seasonality e.g., if authors believe COVID-19 stringency measures interrupted RSV seasonal patterns, then a term that captures contact stringency scaled between 0 and 1 could multiply the seasonal term such that if the term is 1 then its equivalent to modelling typical season whereas if its 0 then there is total interruption to seasonality (equivalent to what authors have done in this work). This term can be informed by real world longitudinal data of how this value changes during black swan or typical RSV seasons e.g., Oxford Stringency Index (https://ourworldindata.org/covid-stringency-index).

• I think this work is good and adds to a discussion around how to forecast future RSV epidemics after transmission interruption due to non-pharmaceutical interventions or vaccination to equip healthcare services.

 

Title

• Machine learning forecasts for seasonal epidemic peaks: “Black swan seasons” and the dangers of overfitting seasonality. In the entire article, the authors report that models without seasonality during “Black swan seasons” perform better in terms of forecasting, which I think is different from overfitting. Does including a variable (seasonality) in the model constitute overfitting? I suggest removing “Black swan seasons and the dangers of overfitting seasonality” in the title.

Abstract

• Lines 15-16: Should be revised, and “black swan” can simply be bracketed and then defined later and about its originality in the introduction section than in abstract.

• Line 19: Traditional modelling should be defined in the abstract

Methods

• Lines 158-166: Even though authors state they used pilot data, there is little description of how that data is generated. Could authors expand how reporting cough is related to RSV etc.

Discussion

• Lines 251-260 seem to be repeating results and not discussing results in the context of other related studies which may have forecasted peak, onset etc. using different methods. Could authors either delete this or discuss it in context.

• As suggested in the Summary, in lines 271-275, authors should consider discussing alternative approach of having seasonality as intrinsic part of seasonal RSV dynamics but including a term that alters seasonality during typical or black swan events

• Could authors cite lines 280-282

Ethics

• My understanding is that the study used pilot human data on cough as proxy for RSV and should therefore state any approval that allowed conduct of the study.

Data/code availability

• The link provided by authors for data availability does not work and was unable to look at the data

• The link to reproducible analysis script not available and was unable to look at the analysis code

Figures

• Figures presented in the manuscript do not have caption and are currently hard to interpret.

• I thought the last three figures could be combined horizontally as a single figure to easily compare across

• Also, the interpretation of the violin plots could be stated clearly in the caption of the figure as it may not be obvious to readers to interpret it.

6. PLOS authors have the option to publish the peer review history of their article (what does this mean?). If published, this will include your full peer review and any attached files.

Reviewer #1: No

---

## [Author Response · Author response to Decision Letter 0]

11 Aug 2023

The authors explore different types of models that forecast epidemic peaks such as of RSV during typical RSV season and out-of-season (black swan) which may have been driven by COVID-19 stringency measures that also reduced contacts among individuals during typical RSV season. They train different models (with or without different terms including seasonality) and test their performance on test data which include that during 2020/2021 RSV season and its delayed epidemic, and compute and compare forecasts errors to identify models that perform better. Overall, I think the paper is well written and could be considered for publication after minor collections.

• A major point is my alternative idea to authors’ line of reasoning; I think seasonality is intrinsic to RSV dynamics or any kind of seasonal respiratory infection and suggesting that models without seasonality could perform better during black swan seasons may be misleading (though true). An alternative approach would be to still maintain the seasonal term in RSV models while including a term that alters normal seasonality e.g., if authors believe COVID-19 stringency measures interrupted RSV seasonal patterns, then a term that captures contact stringency scaled between 0 and 1 could multiply the seasonal term such that if the term is 1 then its equivalent to modelling typical season whereas if its 0 then there is total interruption to seasonality (equivalent to what authors have done in this work). This term can be informed by real world longitudinal data of how this value changes during black swan or typical RSV seasons e.g., Oxford Stringency Index (https://ourworldindata.org/covid-stringency-index).

• I think this work is good and adds to a discussion around how to forecast future RSV epidemics after transmission interruption due to non-pharmaceutical interventions or vaccination to equip healthcare services.

 

We are grateful for the reviewers comments and constructive suggestions. In particular, we like the idea of scaling the seasonality aspects of a model as a potential solution for adapting baselines during a major incident like the Covid-19 pandemic. This is an idea we may pursue further for ongoing surveillance activities.

Title

• Machine learning forecasts for seasonal epidemic peaks: “Black swan seasons” and the dangers of overfitting seasonality. In the entire article, the authors report that models without seasonality during “Black swan seasons” perform better in terms of forecasting, which I think is different from overfitting. Does including a variable (seasonality) in the model constitute overfitting? I suggest removing “Black swan seasons and the dangers of overfitting seasonality” in the title.

We agree that the term “Black swan seasons” is novel and potentially confusing and that overfitting is probably the wrong term to use here, therefore we have renamed the paper – “Machine learning forecasts for seasonal epidemic peaks: lessons learnt from an atypical respiratory syncytial virus season.” We agree with the reviewer’s earlier comments that RSV does exhibit a strong seasonality and that we need to be clearer about what we mean by overfitting in this context. Therefore, we have removed the reference to overfitting in the abstract. We have also removed or clarified other references to overfitting in lines 60, 85, and 168.

Abstract

• Lines 15-16: Should be revised, and “black swan” can simply be bracketed and then defined later and about its originality in the introduction section than in abstract.

We have moved the definition of ‘black swan season’ from the abstract to the introduction and to avoid any confusion we have removed other references to the term ‘black swan’ from the abstract. 

• Line 19: Traditional modelling should be defined in the abstract

By traditional modelling we meant, simple models that assume a predictable seasonality, we have removed this from the abstract as it might add confusion.

Methods

• Lines 158-166: Even though authors state they used pilot data, there is little description of how that data is generated. Could authors expand how reporting cough is related to RSV etc.

We have added a new reference to a paper showing how NHS 111 cough calls in children aged under 5 years corresponds to RSV outbreak.

Discussion

• Lines 251-260 seem to be repeating results and not discussing results in the context of other related studies which may have forecasted peak, onset etc. using different methods. Could authors either delete this or discuss it in context.

We thank the reviewer for this observation - we have deleted the paragraph from the discussion.

• As suggested in the Summary, in lines 271-275, authors should consider discussing alternative approach of having seasonality as intrinsic part of seasonal RSV dynamics but including a term that alters seasonality during typical or black swan events

We thank the reviewer for this suggestion and have added some additional text to the discussion.

• Could authors cite lines 280-282

We have added a new reference for cross-validation.

Ethics

• My understanding is that the study used pilot human data on cough as proxy for RSV and should therefore state any approval that allowed conduct of the study.

We used anonymised health service data that is routinely used by UKHSA for public health surveillance of respiratory illnesses, including RSV. This study was part of ongoing work to improve the capabilities of UKHSA surveillance systems. As such, no specific approvals were required to use the anonymised data included this study. 

Data/code availability

• The link provided by authors for data availability does not work and was unable to look at the data

• The link to reproducible analysis script not available and was unable to look at the analysis code

We are in the process of making the data and scripts available on request.

Figures

• Figures presented in the manuscript do not have caption and are currently hard to interpret.

Figure captions are included in the body of the text. We have added some further description to the captions to aid interpretation. 

• I thought the last three figures could be combined horizontally as a single figure to easily compare across

We felt that the figures would be harder to see if made smaller and combined into a single figure.

• Also, the interpretation of the violin plots could be stated clearly in the caption of the figure as it may not be obvious to readers to interpret it.

We have included a definition of violin plot to the caption for figure 1.

---

## [Editor Report · Decision Letter 1]

11 Sep 2023

Machine learning forecasts for seasonal epidemic peaks: lessons learnt from an atypical respiratory syncytial virus season

PONE-D-22-35171R1

Dear Dr. Morbey,

We’re pleased to inform you that your manuscript has been judged scientifically suitable for publication and will be formally accepted for publication once it meets all outstanding technical requirements.

Kind regards,

André Ricardo Ribas Freitas

Academic Editor

PLOS ONE

---

## [Editor Report · Acceptance letter]

14 Sep 2023

PONE-D-22-35171R1 

Machine learning forecasts for seasonal epidemic peaks: lessons learnt from an atypical respiratory syncytial virus season 

Dear Dr. Morbey:

I'm pleased to inform you that your manuscript has been deemed suitable for publication in PLOS ONE. Congratulations! Your manuscript is now with our production department. 

Kind regards, 

on behalf of

Dr. André Ricardo Ribas Freitas 

Academic Editor

PLOS ONE